# ABSOLUTE POLICY OPTIMIZATION

## ABSTRACT

In recent years, trust region on-policy reinforcement learning has achieved impressive results in addressing complex control tasks and gaming scenarios. However, contemporary state-of-the-art algorithms within this category primarily emphasize improvement in expected performance, lacking the ability to control over the worst-case performance outcomes. To address this limitation, we introduce a novel objective function; optimizing which leads to guaranteed monotonic improvement in the lower bound of near-total performance samples. We call it improvement of absolute performance. Building upon this groundbreaking theoretical advancement, we further introduce a practical solution called Absolute Policy Optimization (APO). Our experiments demonstrate the effectiveness of our approach across challenging continuous control benchmark tasks and extend its applicability to mastering Atari games. Our findings reveal that APO significantly outperforms state-of-the-art policy gradient algorithms, resulting in substantial improvements in worst-case performance, as well as expected performance.

## 1 INTRODUCTION

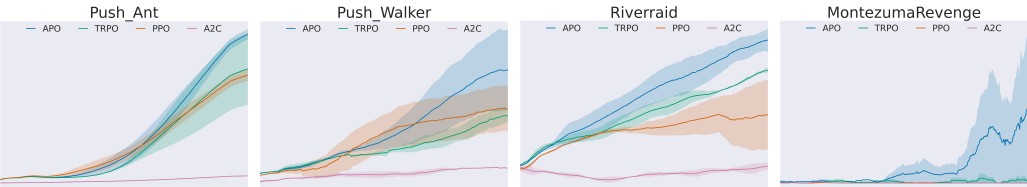

Figure 1: Illustration of performance improvement.

Existing reinforcement learning algorithms have taken the expectation of improving cumulative rewards (referred to as **performance**) as their core optimization objective. Within this framework, trust region-based on-policy reinforcement learning algorithms have achieved the most outstanding results. However, the representative trust-region policy optimization (TRPO) Schulman et al. (2015) only ensures the monotonic improvement of the expectation of this performance distribution, it fails to exert control over the worst-case performance sample originating from the same distribution. In this paper, we introduce a novel theoretical breakthrough that ensures the monotonic improvement of the lower bound of near-total performance samples (**absolute performance**) from the distribution. Subsequently, we implement a series of approximations to transform this theoretically-grounded algorithm into a practical solution, which we refer to as **A**bsolute **P**olicy **O**ptimization (APO). Remarkably, APO exhibits scalability and can efficiently optimize nonlinear policies characterized by tens of thousands of parameters. Our experimental results underscore the effectiveness of APO, demonstrating substantial performance improvements in terms of both absolute performance and expected performance compared to state-of-the-art policy gradient algorithms. These improvements are evident across challenging continuous control benchmark tasks and extend to the realm of playing Atari games. Figure 1 provides a visual representation of our approach's superiority, where APO can effectively handle tasks that other algorithms inherently struggle to optimize, spanning both continuous and discrete control domains. This work serves as the next generation base reinforcement learning algorithm, and represents a significant step towards developing practical RL algorithms that can be robustly applied to many real-world problems.

## 2 RELATED WORKS

**Model-Free Deep Reinforcement Learning**    Model-free deep reinforcement learning (RL) algorithms have found applications from the realm of games (Mnih et al., 2013b; Silver et al., 2016) to the intricate domain of robotic control (Schulman et al., 2015). The leading contenders of the model free reinforcement learning algorithms include (i) deep Q-learning (Mnih et al., 2013a; Hausknecht & Stone, 2015; van Hasselt et al., 2015; Hessel et al., 2018), (ii) off-policy policy gradient methods (Silver et al., 2014; Lillicrap et al., 2015; Gu et al., 2016; Fujimoto et al., 2018; Haarnoja et al., 2018), and (iii) trust region on-policy policy gradient methods (Schulman et al., 2015; 2017).

Among those categories, Q-learning-based techniques, augmented with function approximation, have exhibited remarkable prowess over tasks with discrete action spaces, e.g. Atari game playing (Bellemare et al., 2013). However, these methods performs poorly in the realm of continuous control benchmarks, notably exemplified in OpenAI Gym (Brockman et al., 2016a; Duan et al., 2016).

In contrast, off-policy policy gradient methods extend Q-learning-based strategies via introducing an independent actor network to handle continuous control tasks, as exemplified by the Deep Deterministic Policy Gradient (DDPG)(Lillicrap et al., 2015). However, off-policy methods suffer from stability issues and susceptibility to hyperparameter tuning nuances(Haarnoja et al., 2018). Recently, enhancements are made to incorporate entropy to foster exploration (Haarnoja et al., 2018) and mitigate the overestimation bias through target networks (Fujimoto et al., 2018). Despite these advancements, the convergence characteristics of off-policy policy gradient methods remain incompletely understood, primarily explored under stringent assumptions such as infinite sampling and Q-updates (Fujimoto et al., 2018). Moreover, off-policy policy gradient methods are primarily tailored for continuous action spaces.

Conversely, trust region on-policy policy gradient methods harmoniously accommodate both continuous and discrete action spaces while showcasing superior stability and dependable convergence properties. Notably, the representative Trust Region Policy Optimization (TRPO)Schulman et al. (2015), complemented by its pragmatic counterpart, Proximal Policy Optimization (PPO)Schulman et al. (2017), have consistently delivered impressive performance across an array of demanding benchmark tasks. Furthermore, those methods have largely helped training of groundbreaking artificial intelligence applications, including ChatGPT (Schulman et al., 2022), the automated Rubik's Cube solver with a robotic hand (Akkaya et al., 2019), and the championship-level drone racing (Kaufmann et al., 2023), thereby reaffirming their profound impact on advancing the frontiers of AI technology.

**Attempts to Improve Trust Region Methods**    Recently, many efforts are made to improve trust region on-policy methods, including (i) *improve computation efficiency*. TREFree  (Sun et al., 2023) introduced a novel surrogate objective that eliminates trust region constraints. (ii) *encourage exploration*. COPOS (Pajarinen et al., 2019) applied compatible value function approximation to effectively control entropy during policy updates. (iii) *improve training stability and data-efficiency*. Truly PPO (TR-PPO) (Wang et al., 2020) introduced a new clipping function and trust region-based triggering condition. Generalized PPO (GePPO)  (Queeney et al., 2021) extended PPO to an off-policy variant, thereby enhancing sampling efficiency through data reuse. AlphaPPO  (Xu et al., 2023) introduced alpha divergence, a parametric metric that offers a more effective description of policy differences, resulting more stable training performance.

There are also improvements considering variance control, including (i) *variance reduction of policy gradient*.  Xu et al. and Papini et al. applied the stochastic variance reduced gradient descent (SVRG) technique for getting stochastic variance-reduced version of policy gradient (SVRPO) to improve the sample efficiency. (Yuan et al.) incorporates the StochAstic Recursive grAdient algoritHm (SARAH) into the TRPO framework to get more stable variance. (ii) *variance reduction of performance update*. (Tomczak et al., 2019) introduced a surrogate objective with approximate importance sampling to strike a balance between performance update bias and variance. (iii) *variance reduction of importance sampling*. (Lin et al., 2023) introduced sample dropout to bound the variance of importance sampling estimate by dropping out samples when their ratio deviation is too high.

Although trust region-based methods have achieved notable success, there remains substantial potential for improvement. A critical gap in existing approaches lies in their inability to exert control over the worst-case individual performance samples stemming from the policy. Unforeseen instances of poor performance can result in training instability, thereby jeopardizing the reliability of solutions

in real-world applications. In our research, we bridge this gap by introducing novel theoretical results that ensure a monotonic improvement of the lower bound of near-total performance samples.

# 3 PROBLEM FORMULATION

## 3.1 NOTATIONS

Consider an infinite-horizon discounted Markov decision process (MDP) defined by the tuple $(\mathcal{S}, \mathcal{A}, \gamma, \mathcal{R}, P, \mu)$, where $\mathcal{S}$ is the state space, and $\mathcal{A}$ is the control space, $R : \mathcal{S} \times \mathcal{A} \mapsto \mathbb{R}$ is a bounded reward function, $0 \leq \gamma < 1$ is the discount factor, $\mu : \mathcal{S} \mapsto \mathbb{R}$ is the bounded initial state distribution, and $P : \mathcal{S} \times \mathcal{A} \times \mathcal{S} \mapsto \mathbb{R}$ is the transition probability. $P(s'|s, a)$ is the probability of transitioning to state $s'$ when the agent takes action $a$ at state $s$. A stationary policy $\pi : \mathcal{S} \mapsto \mathcal{P}(\mathcal{A})$ is a mapping from states to a probability distribution over actions, with $\pi(a|s)$ denoting the probability of selecting action $a$ in state $s$. We denote the set of all stationary policies by $\Pi$. Subsequently, we denote $\pi_\theta$ as the policy that is parameterized by the parameter $\theta$.

The standard goal for MDP is to learn a policy $\pi$ that maximizes a performance measure $\mathcal{J}(\pi)$ which is computed via the discounted sum of reward:

$$\mathcal{J}(\pi) = \mathbb{E}_{\tau \sim \pi} \left[ \sum_{t=0}^{\infty} \gamma^t R(s_t, a_t, s_{t+1}) \right], \tag{1}$$

where $\tau = [s_0, a_0, s_1, \cdots]$, and $\tau \sim \pi$ is shorthand for that the distribution over trajectories depends on $\pi : s_0 \sim \mu, a_t \sim \pi(\cdot|s_t), s_{t+1} \sim P(\cdot|s_t, a_t)$. And the objective is to select a policy $\pi$ that maximizes the performance measure: $\max\limits_{\pi \in \Pi} \mathcal{J}(\pi)$.

## 3.2 ABSOLUTE PERFORMANCE BOUND

Notice that the above considers maximizing the expected reward performance, which, unfortunately, does not provide control over each individual performance sample derived from the policy $\pi$. To clarify, a performance sample is defined here as $R_\pi(s_0) \doteq \sum_{t=0}^{\infty} \gamma^t R(s_t, a_t, s_{t+1})$, where the state action sequence $\hat{\tau} = [a_0, s_1, \ldots] \sim \pi$ starts with an initial state $s_0$, which follows initial state distribution $\mu$. In practical reinforcement learning setting, unexpected poor performance samples can lead to training instability, compromising the reliability of solutions in real-world applications. To tackle this issue, our fundamental insight is that policy optimization should not be solely fixated on enhancing expected performance, but also on improving the worst-case performance samples originating from the distribution of the variable $R_\pi(s_0)$.

However, it's important to acknowledge that within the Markov Decision Process framework, any possible state visitation is inherently assigned a non-zero probability. In essence, any policy performance distribution inherently accommodates the statistical possibility of all conceivable $R_\pi(s_0)$. Therefore, our ultimate goal is to improve the lower bound of near-total performance samples derived from the policy. We denote this lower bound as the absolute performance bound for the policy:

**Definition 1** (Absolute Performance Bound). *$\mathcal{B}_k(\pi)$ is called the absolute performance bound with $p_k$ confidence if it satisfies the following condition:*

$$Pr\big(R_\pi(s_0) \geq \mathcal{B}_k(\pi)\big) \geq p_k, \tag{2}$$

*where $p_k \doteq 1 - \frac{1}{k^2} \in (0, 1)$ and $k$ is the probability factor ($k > 1$ and $k \in \mathbb{R}$), which can be set according to the demand for probability magnitude.*

**Remark 1.** *Definition 1 shows that more than $p_k$ of the samples from the distribution of $R_\pi(s_0)$ will be larger than the bound $\mathcal{B}_k(\pi)$. By setting $p_k \to 1$, $\mathcal{B}_k(\pi)$ represents the lower bound of near-total performance samples of policy $\pi$.*

## 3.3 ABSOLUTE MARKOV DECISION PROCESS

In this paper, our focus is on a special class of Markov Decision Processes (MDP) characterized by improvement of the absolute absolute performance. This unique class is termed **A**bsolute **M**arkov **D**ecision **P**rocess (**AMDP**). Much like a standard MDP, an AMDP is defined by a tuple $(\mathcal{S}, \mathcal{A}, \gamma, \mathcal{R}, P, \mu, k)$, with the inclusion of an extra probabilistic factor denoted as $k$, which

serves to modulate the degree of conservatism in the absolute performance bound. In accordance with Definition 1, the overarching objective within the AMDP framework is to identify a policy $\pi$ that effectively improves $\mathcal{B}_k$. For an unknown performance distribution, we first define $\mathcal{V}(\pi) \doteq \mathbb{E}_{\tau \sim \pi}\left[\left(\sum_{t=0}^{\infty} \gamma^t R(s_t, a_t, s_{t+1}) - \mathcal{J}(\pi)\right)^2\right]$ as the variance of the performance distribution. Then, we can leverage the Chebyshev's inequality theory Saw et al. (1984) to obtain an absolute performance bound as $\mathcal{B}_k(\pi) \doteq \mathcal{J}(\pi) - k\mathcal{V}(\pi)$, which is guaranteed to satisfy Definition 1 (proved in Proposition 1). Thus, AMDP addresses the following optimization

$$\max_{\pi \in \Pi} \mathcal{J}(\pi) - k\mathcal{V}(\pi). \tag{3}$$

Here we define the on-policy value function as $V_\pi(s) \doteq \mathbb{E}_{\tau \sim \pi}[R_\pi(s)|s_0 = s]$, the on-policy action-value function as $Q_\pi(s, a) = Q_\pi(s, a, s') \doteq \mathbb{E}_{\tau \sim \pi}[R_\pi(s)|s_0 = s, a_0 = a]$, and the advantage function as $A_\pi(s, a) = A_\pi(s, a, s') \doteq Q_\pi(s, a) - V_\pi(s)$. We also define $\bar{A}_\pi(s)$ as the expected advantage of $\pi'$ over $\pi$ at state $s$: $\bar{A}_\pi(s) \doteq \mathbb{E}_{a \sim \pi'}[A_\pi(s, a)]$.

## 4 ABSOLUTE POLICY OPTIMIZATION

To optimize equation 3, we need to evaluate the objective with respect to an unknown $\pi$. Our main intuition is to find a surrogate function for the objective, such that (i) it represents a tight lower bound of the objective; and (ii) it can be easily estimated from the samples on the most recent policy. To solve large and continuous AMDPs, policy search algorithms look for the optimal policy within a set $\Pi_\theta \subset \Pi$ of parametrized policies. Mathematically, APO updates solve the following optimization:

$$\pi_{j+1} = \underset{\pi \in \Pi_\theta}{\text{argmax}} \, \mathcal{J}^l_{\pi,\pi_j} - k\left(MV_{\pi,\pi_j} + VM_{\pi,\pi_j}\right) \quad, \tag{4}$$

where $\mathcal{J}^l_{\pi,\pi_j}$ represents the lower bound surrogate function for $\mathcal{J}(\pi)$ and $\left(MV_{\pi,\pi_j} + VM_{\pi,\pi_j}\right)$ represents the upper bound surrogate function for $\mathcal{V}(\pi)$.

**Remark 2.** *Since the performance samples from the same start state belong to a one-dimensional distribution, performance samples from different start states belong to a mixture of one-dimensional distributions. $MV_{\pi,\pi_j}$ reflects the upper bound of expected variance of the return over different start states. $VM_{\pi,\pi_j}$ reflects the upper bound of variance of the expected return of different start states. The detailed interpretations are discussed in Equation (18), Lemma 1, and Lemma 2.*

Here $\mathcal{J}^l_{\pi,\pi_j}, MV_{\pi,\pi_j}, VM_{\pi,\pi_j}$ are defined as:

$$\mathcal{J}^l_{\pi,\pi_j} \doteq \mathcal{J}(\pi_j) + \frac{1}{1-\gamma}\mathbb{E}_{s \sim d^{\pi_j}, a \sim \pi}\left[A_{\pi_j}(s, a) - \frac{2\gamma\epsilon^\pi}{1-\gamma}\sqrt{\frac{1}{2}\mathcal{D}_{KL}(\pi\|\pi_j)[s]}\right] \tag{5}$$

$$MV_{\pi,\pi_j} \doteq \frac{\|\mu^T\|_\infty}{1-\gamma^2}\max_s \left| \mathbb{E}_{\substack{a \sim \pi \\ s' \sim P}}\left[A_{\pi_j}(s, a, s')^2\right] - \mathbb{E}_{\substack{a \sim \pi_j \\ s' \sim P}}\left[A_{\pi_j}(s, a, s')^2\right] + |H(s, a, s')|^2_{max} \right. \tag{6}$$

$$\left. + 2\mathbb{E}_{\substack{a \sim \pi \\ s' \sim P}}\left[A_{\pi_j}(s, a, s')\right] \cdot |H(s, a, s')|_{max} \right| + MV_{\pi_j} + \frac{2\gamma^2\|\mu^T\|_\infty}{1-\gamma^2}\sqrt{\frac{1}{2}\mathcal{D}_{KL}(\pi\|\pi_j)[s]} \cdot \|\Omega_{\pi_j}\|_\infty$$

$$VM_{\pi,\pi_j} \doteq \|\mu^T\|_\infty \max_s \left| |\eta(s)|^2_{max} + 2|V_{\pi_j}(s)| \cdot |\eta(s)|_{max} \right| - \min \left(\mathcal{J}(\pi)\right)^2 + \mathbb{E}_{s_0 \sim \mu}[V^2_{\pi_j}(s_0)] \tag{7}$$

where $\mathcal{D}_{KL}(\pi\|\pi_j)[s]$ is KL divergence between two policy $(\pi, \pi_j)$ at state $s$, $\epsilon^\pi \doteq \max_s|\mathbb{E}_{a \sim \pi}[A_{\pi_j}(s, a)]|$, $d^{\pi_j} \doteq (1-\gamma)\sum_{t=0}^{H} \gamma^t P(s_t = s|\pi_j)$, $\Omega_{\pi_j} \doteq \begin{bmatrix} \omega_{\pi_j}(s^1) \\ \omega_{\pi_j}(s^2) \\ \vdots \end{bmatrix}$, $\omega_{\pi_j}(s) \doteq \mathbb{E}_{a \sim \pi_j, s' \sim P}[Q_{\pi_j}(s, a, s')^2] - V_{\pi_j}(s)^2$, $MV_{\pi_j} \doteq \mathbb{E}_{s_0 \sim \mu}[\mathbb{V}ar[R_{\pi_j}(s_0)]$ and $\min \left(\mathcal{J}(\pi)\right)^2 \doteq \min_{\mathcal{J}(\pi) \in [\mathcal{J}^l_{\pi,\pi_j}, \mathcal{J}^u_{\pi,\pi_j}]} \left(\mathcal{J}(\pi)\right)^2$ with $\mathcal{J}^u_{\pi,\pi_j} \doteq \mathcal{J}(\pi_j) + \frac{1}{1-\gamma}\mathbb{E}_{\substack{s \sim d^{\pi_j} \\ a \sim \pi}}\left[A_{\pi_j}(s, a) + \frac{2\gamma\epsilon^\pi}{1-\gamma}\sqrt{\frac{1}{2}\mathcal{D}_{KL}(\pi\|\pi_j)[s]}\right]$.

Additionally,

$$|H(s,a,s')|_{max} \doteq \left| \gamma \underset{\substack{s_0=s' \\ \hat{\tau} \sim \pi_j}}{\mathbb{E}} \left[ \sum_{t=0}^{\infty} \gamma^t \bar{A}_{\pi_j}(s_t) \right] - \underset{\substack{s_0=s \\ \hat{\tau} \sim \pi_j}}{\mathbb{E}} \left[ \sum_{t=0}^{\infty} \gamma^t \bar{A}_{\pi_j}(s_t) \right] \right| + \frac{2\gamma(1+\gamma)\epsilon}{(1-\gamma)^2} \mathcal{D}_{KL}(\pi||\pi_j)[s] \tag{8}$$

$$|\eta(s)|_{max} \doteq \left| \underset{\substack{s_0=s \\ \hat{\tau} \sim \pi_j}}{\mathbb{E}} \left[ \sum_{t=0}^{\infty} \gamma^t \bar{A}_{\pi_j}(s_t) \right] \right| + \frac{2\gamma\epsilon}{(1-\gamma)^2} \mathcal{D}_{KL}(\pi||\pi_j)[s] \tag{9}$$

$$\epsilon \doteq \mathbf{max}_{s,a}|A_{\pi_j}(s,a)| \ . \tag{10}$$

**Theoretical Guarantees for APO**

**Theorem 1** (Monotonic Improvement of Absolute Performance). *Suppose $\pi, \pi'$ are related by equation 4, then absolute performance bound $\mathcal{B}_k(\pi) = \mathcal{J}(\pi) - k\mathcal{V}(\pi)$ satisfies $\mathcal{B}_k(\pi') \geq \mathcal{B}_k(\pi)$.*

The proof for Theorem 1 is summarized in Appendix A.

## 5 OPTIMIZED APO

In addition to the guaranteed enhancement of the absolute performance bound, it is also highly desirable for policy optimization to improve expected performance. By leveraging the concept of policy improvement bounds as introduced in trust region methods (Schulman et al., 2015; Achiam et al., 2017), we can enhance APO by imposing additional constraint on the feasible solutions:

$$\pi_{j+1} = \underset{\pi \in \Pi_\theta}{\mathbf{argmax}} \ \mathcal{J}^l_{\pi,\pi_j} - k\left(MV_{\pi,\pi_j} + VM_{\pi,\pi_j}\right) \ \mathbf{s.t.} \ \mathcal{J}^l_{\pi,\pi_j} \geq \mathcal{J}(\pi_j) \tag{11}$$

**Remark 3.** *Any feasible solution to equation 11 ensures (i) monotonic improvement of absolute performance [Theorem 1], and (ii) monotonic improvement of expected performance [Corollary 1, Corollary 3, (Achiam et al., 2017)]. Additionally, $\pi_j$ is always a feasible solution towards equation 11.*

In local policy search (Peters & Schaal, 2008), the policy is iteratively updated by maximizing objective within a local neighborhood of the most recent policy $\pi_j$. Inspired by trust region methods, we approximates equation 11 with a trust region constraint instead of penalties on policy divergence:

$$\pi_{j+1} = \underset{\pi \in \Pi_\theta}{\mathbf{argmax}} \ \frac{1}{1-\gamma} \underset{\substack{s \sim d^{\pi_j} \\ a \sim \pi}}{\mathbb{E}} \left[ A_{\pi_j}(s,a) \right] - k\left(\overline{MV}_{\pi,\pi_j} + \overline{VM}_{\pi,\pi_j}\right) \tag{12}$$

$$\mathbf{s.t.} \ \frac{1}{1-\gamma} \underset{\substack{s \sim d^{\pi_j} \\ a \sim \pi}}{\mathbb{E}} \left[ A_{\pi_j}(s,a) \right] \geq 0 \ \ \mathrm{and} \ \ \bar{\mathcal{D}}_{KL}(\pi||\pi_j) \leq \delta$$

where $\delta$ is the step size, $\overline{MV}_{\pi,\pi_j} \doteq MV_{\pi,\pi_j} - MV_{\pi_j} - \frac{2\gamma^2||\mu^T||_\infty}{1-\gamma^2}\sqrt{\frac{1}{2}\mathcal{D}_{KL}(\pi||\pi_j)[s]} \cdot ||\Omega_{\pi_j}||_\infty$ and $\overline{VM}_{\pi,\pi_j} = VM_{\pi,\pi_j} - \underset{s_0 \sim \mu}{\mathbb{E}}[V^2_{\pi_j}(s_0)]$. The set $\{\pi \in \Pi_\theta : \bar{\mathcal{D}}_{KL}(\pi||\pi_j) = \mathbb{E}_{s \sim \pi_j}[\mathcal{D}_{KL}(\pi||\pi_j)[s]] \leq \delta\}$ is called *trust region*. Notice that $MV_{\pi_j}$ and $\mathbb{E}_{s_0 \sim \mu}[V^2_{\pi_j}(s_0)]$ are computable constant.

Equation 12 represents a constrained policy optimization problem, the solution to which has been discussed in prior work (Achiam et al., 2017). In practical implementations, due to various approximation errors, equation 12 may often lead to infeasible solutions. Hence, the line search trick (Achiam et al., 2017) results in very small step sizes and makes it hard to update the policy effectively. To overcome that challenge, we optimize the problem in the form of weighted sum:

$$\pi_{j+1} = \underset{\pi \in \Pi_\theta}{\mathbf{argmax}} \ \left( w_1 \left( \frac{1}{1-\gamma} \underset{\substack{s \sim d^{\pi_j} \\ a \sim \pi}}{\mathbb{E}} \left[ A_{\pi_j}(s,a) \right] \right) + \tag{13}$$

$$w_2 \left( \frac{1}{1-\gamma} \underset{\substack{s \sim d^{\pi_j} \\ a \sim \pi}}{\mathbb{E}} \left[ A_{\pi_j}(s,a) \right] - k\left(\overline{MV}_{\pi,\pi_j} + \overline{VM}_{\pi,\pi_j}\right) \right) \right)$$

$$\mathbf{s.t.} \ \bar{\mathcal{D}}_{KL}(\pi||\pi_j) \leq \delta$$

where $w_1, w_1$ are weights to control the importance of optimization objectives.

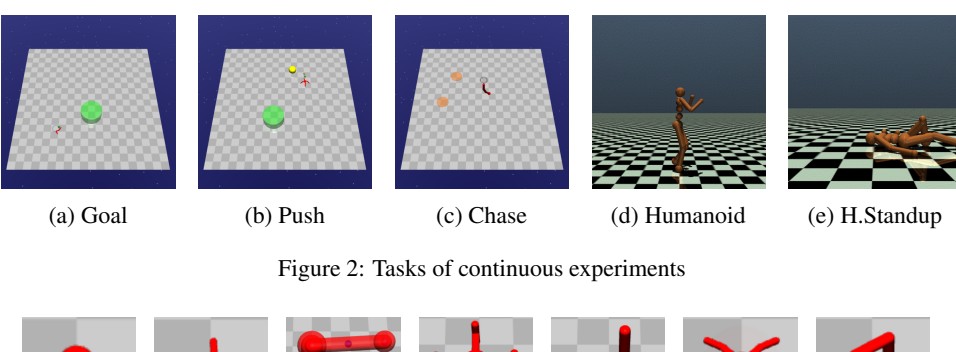

Figure 2: Tasks of continuous experiments

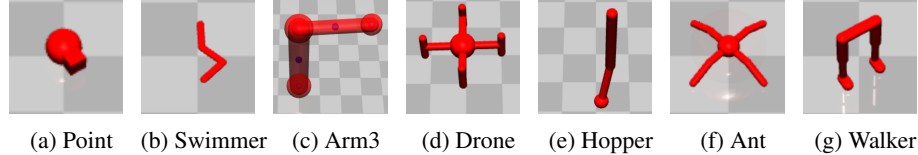

Figure 3: Robots of continuous tasks benchmark GUARD.

## 6 PRACTICAL IMPLEMENTATION

In this section, we show how to (i) simplify complex computations (ii) implement an efficient approximation to Equation (13) and (iii) encourage learning even when Equation (13) becomes infeasible. The full APO pseudocode is provided as Algorithm 1 in Appendix B.

**Special Hyperparameters for Practical Implementation** For practical implementation, we treat two items when implementing Equation (13) as hyperparameters. (i) $\|\boldsymbol{\mu^T}\|_{\boldsymbol{\infty}}$: we treat the infinity norm of $\mu$ (initial distribution of the system) as a constant parameter due to its inaccessibility during implementation. (ii) $|\boldsymbol{H(s,a,s')}|_{\boldsymbol{max}}$: We can either compute $|H(s,a,s')|_{max}$ from the most recent policy with equation 8 or treat it as a hyperparameter since $|H(s,a,s')|_{max}$ is bounded for any system with a bounded reward function. In practice, we found that the hyperparameter option helps increasing the performance. This setting will be discussed more detailedly in Section 7.5.

**Efficient Approximation of APO** We first estimate the expected reward advantage in Equation (13) via importance sampling with a sampling distribution $\pi_k$ (Schulman et al., 2015) as

$$\mathbb{E}_{a\sim\pi,s'\sim P}\left[A_{\pi_j}(s,a,s')^2\right] = \mathbb{E}_{a\sim\pi_j,s'\sim P}\left[(\pi(a|s)/\pi_j(a|s))\,A_{\pi_j}(s,a,s')^2\right] \tag{14}$$

Equation (14) allows us to replace $\mathbb{E}_{a\sim\pi,s'\sim P}\left[A_{\pi_j}(s,a,s')^2\right]$ with empirical estimates at each state-action pair $(s,a)$ from rollouts by the previous policy $\pi_j$. The empirical estimate of reward advantage is given by $R(s,a,s') + \gamma V_{\pi_j}(s') - V_{\pi_j}(s)$. $V_{\pi_j}(s)$ can be computed at each state by taking the discounted future return. To proceed, we convexify Equation (13) by approximating the objective via first-order expansions, and the trust region constraint via second-order expansions. Then, Equation (13) can be efficiently solved using duality (Schulman et al., 2015).

**Infeasible Solution** An update to $\theta$ is computed according to the techniques described in Schulman et al. (2015) every time Equation (13) is solved. However, due to approximation errors, sometimes Equation (13) can become infeasible. In that case, we use a line search to ensure improvement of the surrogate objective and satisfaction of the KL divergence constraint. Starting with the maximal value of the step length, we shrink it exponentially until the objective improves. Without this line search, the algorithm occasionally computes large steps that cause a catastrophic degradation of performance.

## 7 EXPERIMENT

In our experiements, we want to answer the following questions:
**Q1:** How does APO compare with state-of-the-art on-policy RL algorithms?
**Q2:** What benefits are demonstrated by directly optimizing the absolute performance?
**Q3:** Is treating $H_{max}$ as a hyparameter necessary?
**Q4:** What are the impacts of different probability factor $k$ choices?

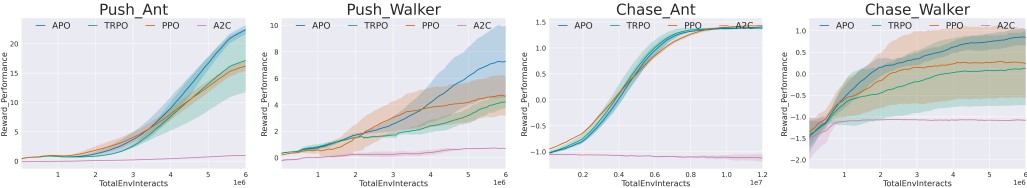

Figure 4: Comparison of results from four representative test suites in low dimensional continuous systems

Figure 5: Comparison of results from four representative test suites in high dimensional continuous systems

## 7.1 EXPERIMENT SETUP

To answer the above, we run experiments on both continuous domain and the discrete domain.

**Continuous Tasks**   Our continuous experiments are conducted on GUARD (Zhao et al., 2023), a challenging robot locomotion benchmark build upon Mujoco (Todorov et al., 2012) and Gym. Seven different robots are included: (i) **Point:**  (Figure 3a) A point-mass robot ($\mathcal{A} \subseteq \mathbb{R}^2$) that can move on the ground. (ii) **Swimmer:**  (Figure 3b) A three-link robot ($\mathcal{A} \subseteq \mathbb{R}^2$) that can move on the ground. (iii) **Arm3:**  (Figure 3c) A fixed three-joint robot arm($\mathcal{A} \subseteq \mathbb{R}^3$) that can move its end effector around with high flexibility. (iv) **Drone:**  (Figure 3d) A quadrotor robot ($\mathcal{A} \subseteq \mathbb{R}^4$) that can move in the air. (v) **Hopper:**  (Figure 3e) A one-legged robot ($\mathcal{A} \subseteq \mathbb{R}^5$) that can move on the ground. (vi) **Ant:**  (Figure 3f) A quadrupedal robot ($\mathcal{A} \subseteq \mathbb{R}^8$) that can move on the ground. (vii) **Walker:** (Figure 3g) A bipedal robot ($\mathcal{A} \subseteq \mathbb{R}^{10}$) that can move on the ground. Furthermore, three different types of tasks are considered, including (i) **Goal:**  (Figure 2a) robot navigates towards a series of 2D or 3D goal positions. (ii) **Push:**  (Figure 2b) robot pushes a ball toward different goal positions. (iii) **Chase:**  (Figure 2c) robot tracks multiple dynamic targets. Considering these different robots and tasks, we design 8 low-dim test suites and 4 high-dim test suits with 7 types of robots and 3 types of tasks, which are summarized in Table 3 in Appendix. We name these test suites as {Task Type}_{Robot}. Further details are listed in Appendix C.1.

Additionally, we conduct continuous control experiments on Mujoco Openai Gym (Brockman et al., 2016b). Two tasks are considered: (i) **Humanoid:**  (Figure 2d) The 3D bipedal robot ($\mathcal{A} \subseteq \mathbb{R}^{17}$) is designed to simulate a human. And the goal of the environment is to walk forward as fast as possible without falling over. (ii) **Humanoid Standup:**  (Figure 2e) The robot ($\mathcal{A} \subseteq \mathbb{R}^{17}$) is same with task **Humanoid**, but the goal is to make the humanoid standup and then keep it standing by applying torques on the various hinges. These two tasks are also summarized in Table 3.

**Discrete Tasks**   We also test APO in all 62 Atari environments of (Brockman et al., 2016b) which are simulated on the Arcade Learning Environment benchmark (Bellemare et al., 2018). All experiments are based on 'v5' environments and 'ram' observation space.

**Comparison Group**   We compare APO to the state-of-the-art base on-policy RL algorithms: (i) TRPO (Schulman et al., 2015) (ii) Advantage Actor Critic (A2C) (Mnih et al., 2016) (iii) PPO (Schulman et al., 2017). For all experiments, we take the best specific parameters mentioned in the original paper and keep the common parameters as the same. The policy $\pi$, the value $V^\pi$ are all encoded in feedforward neural networks using two hidden layers of size (64,64) with tanh activations. The full list of parameters of all methods and tasks compared can be found in Appendix C.2.

## 7.2 COMPARISON TO OTHER ALGORITHMS IN THE CONTINUOUS DOMAIN

### 7.2.1 GUARD BENCHMARK

**Low dimension** Figure 4 shows representative comparison results on low dimensional system (See Appendix D for all results). APO is successful at getting more steady and higher final reward. We notice that PPO only gains faster convergence in part of the simplest task owing to its exploration abilities, the advantage decreases rapidly with more complex tasks such as PUSH. In difficult tasks, APO can perform best at the combined level of convergence speed and final performance.

**High dimension** Figure 5 reports the comparison results on challenging high-dimensional Ant_{PUSH, CHASE} and Walker_{PUSH, CHASE} tasks, where APO outperforms other baselines in getting higher reward and convergence speed.

### 7.2.2 MUJOCO BENCHMARK

To showcase the performance of APO on other high-dimensional continuous benchmark, we conduct additional experiments involving 3D humanoid robot (details have been introduced in Section 7.1) and compare the results with baseline method TRPO. See Table 5 for detailed hyperparameters and Figure 6 for learning curves.

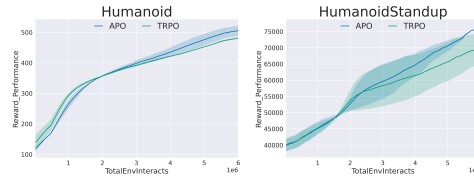

Figure 6: Additional experiments on MuJoCo

## 7.3 COMPARISON TO OTHER ALGORITHMS IN THE ATARI DOMAIN

The hyperparameters for Atari Domain are also provided in Appendix C.2. For the other three algorithms, we used hyperparameters that were tuned to maximize performance on this benchmark. Then we follow the metrics of (Schulman et al., 2017) to quantitatively evaluate the strengths of APO: (i) average expected reward per episode over **all epochs of training** (which favors fast learning), and (ii) average expected reward per episode over **last 100 epochs of training** (which favors final performance). Table 1 records the number of highest evaluation scores obtained by each algorithm across all games. The learning curves for all testing atari games are provided in Appendix D. To compare the performance of all testing algorithm to the TRPO baseline across games, we slightly change the normalization algorithm proposed by (van Hasselt et al., 2015) to obtain more reasonable score (See Appendix C.3 for further explanation of rationality) in percent. The score we used is average reward per episode over last 100 epochs of training:

$$\Delta_1 \doteq score_{agent} - score_{random}, \ \Delta_2 \doteq score_{TRPO} - score_{random} \tag{15}$$

$$score_{normalized} = \frac{\Delta_2}{\Delta_1} \ if \ \Delta_1 < 0 \ and \ \Delta_2 < 0 \ else \ \frac{\Delta_1}{\Delta_2}$$

Then we use stacked bar chart in Figure 7 to visualize APO's capabilities. Figure 7 show that APO has a superior combination of capabilities compared to other algorithms. So far the above experimental comparison answers **Q1**.

|  | APO | PPO | TRPO | A2C | Tie |
|---|---|---|---|---|---|
| (1) average expected reward over all epochs | **26** | 22 | 10 | 3 | 1 |
| (2) average expected reward over last 100 epochs | **29** | 17 | 12 | 3 | 1 |

Table 1: The number of highest evaluation scores obtained by each algorithm across all games

## 7.4 ABSOLUTE PERFORMANCE COMPARISON

We use large probability factor $k$ in practical implementation, which means we are close to optimizing the lower bound for all samples. Thus we use another two similar metrics to evaluate the effectiveness of algorithms for lower bound lifting: (iii) average worst reward per episode over **all epochs of training**, and (iv) average worst reward per episode over **last 20 epochs of training**. We summarize the absolute performance of APO in Atari games and GUARD in Table 2, which answers **Q2**.

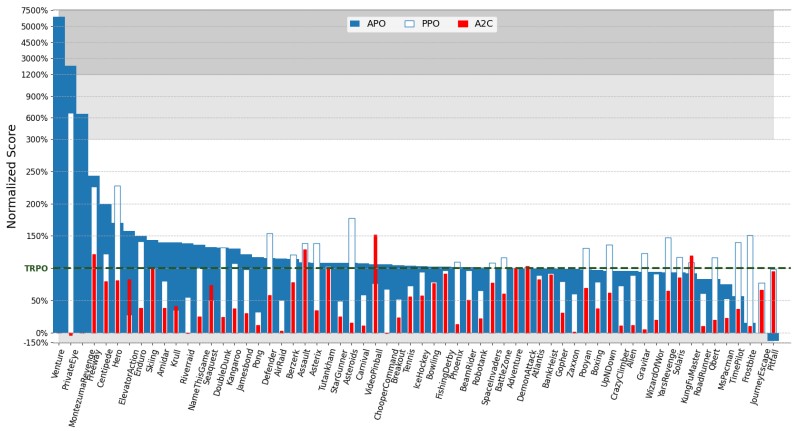

Figure 7: Stacked bar chart for all 62 atari games

|  | APO | PPO | TRPO | A2C | Tie |
|---|---|---|---|---|---|
| (1) average worst reward over all epochs(Atari) | **26** | 24 | 7 | 3 | 2 |
| (2) average worst reward over last 100 epochs(Atari) | **27** | 20 | 10 | 3 | 2 |
| (3) average worst reward over all epochs(GUARD) | **7** | 3 | 1 | 1 | 0 |
| (4) average worst reward over last 20 epochs(GUARD) | **8** | 2 | 2 | 0 | 0 |

Table 2: The number of highest evaluation scores obtained by each algorithm across all games

## 7.5 ABLATION ON $H_{max}$ HYPERPARAMETER TRICK

We chose Riverraid of discrete tasks and PUSH_Ant of continuous tasks to perform ablation experiments against the $|H(s, a, s')|_{max}$ implementation. Figure 8 shows that although both boosts are similar in the early stages of tasks, hypeterparameter method can more consistently converge to a higher reward value. Thus, the figures and description answer **Q3**.

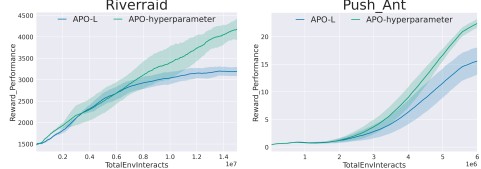

Figure 8: Ablation on $H_{max}$ hyperparameter trick

## 7.6 ABLATION ON PROBABILITY FACTOR $k$

For ablation, we selected Riverraid to investigate the impact of different choices for the probability factor $k$. As illustrated in Figure 9, when $k$ takes on a very small value, indicating optimization of only a limited portion of performance samples, the effectiveness diminishes. This is attributed to the loss of control over the lower bound of near-total performance samples. Conversely, when $k$ becomes excessively large, the optimization shifts its focus towards the most extreme worst-case performance scenarios. This ultra-conservative approach tends to render the overall optimization less effective. Therefore, a moderate choice of $k$ will be favorable to the overall improvement of the effect, which answers **Q4**.

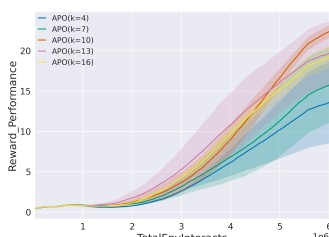

Figure 9: Ablation on probability factor $k$

## 8 CONCLUSION

This paper proposed APO, the first general-purpose policy search algorithm that improve both expected performance and absolute performance. Our approach is grounded by a pioneering theoretical advancement, where maximization of a specific objective function ensures monotonic improvement of expected performance and lower bound of near-total performance samples. We demonstrate APO's effectiveness on challenging continuous control benchmark tasks and playing Atari games, showing its significant performance improvement compared to existing methods and ability to enhance both expected performance and worst-case performance.

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
