# OpenReview forum: "Absolute Policy Optimization"
_ICLR.cc/2024/Conference — ICLR 2024 Conference Withdrawn Submission_

### Official Review · Reviewer_9WdD · 2023-10-14

**Soundness:** 3 good
**Presentation:** 3 good
**Contribution:** 2 fair
**Rating:** 5
**Confidence:** 3

**Summary:**

This paper proposes a novel objective function and a corresponding solution called Absolute Policy Optimization (APO) to address the problem of lacking the ability to control the worst-case outcomes in TRPO-based RL methods. APO can guarantee monotonic improvement in the lower bound of near-total performance samples. Experiments demonstrate the effectiveness of APO by comparing it with TRPO, PPO, and A2C.

**Strengths:**

1. This paper proposes a new objective function and a corresponding solution called APO.
2. The paper is well-written and has good readability.
3. The theoretical analysis in this paper is sufficient

**Weaknesses:**

1. The compared methods include TRPO (2015), PPO (2017), and A2C (2016). I think the recently proposed RL methods (After 2020) should be compared with the proposed one.
2. I suggest the authors improve the recently proposed RL methods by their APO for this comparison.
3. Running time comparison should be given.
4. It seems that a performance sample may be a long-length rollout.

**Questions:**

1. Based on the Markov property of MDPs and Bellman operator, RL can monotonically improve the worst-case outcomes. So what is the advantage of the proposed method? I suggest the authors analyze the advantages of their method on POMDPs, which  do not satisfy the Markov
property,  and give the corresponding experimental results.

2. It seems that AMDPs are a new type of MDPs in this paper.

---

### Official Review · Reviewer_GtPP · 2023-10-31

**Soundness:** 3 good
**Presentation:** 3 good
**Contribution:** 3 good
**Rating:** 5
**Confidence:** 4

**Summary:**

Previous trust-region on-policy RL algorithms mainly focus on improvement in expected performance, while ignoring the control over the worst-case performance outcomes. To address the above limitation, this paper introduces a new objective function, which guarantees monotonic improvement in the lower bound of near-total performance samples. Additionally, this paper proposes a corresponding algorithm named absolute policy optimization and demonstrates the effectiveness of the approach across several continuous control benchmarks.

**Strengths:**

* This paper is well-written and easy to follow.
* The paper is technically sound with most claims supported sufficiently and the theoretical guarantee seems original.
* Extensive experiments are done to verify the effectiveness of the new approach.

**Weaknesses:**

* In Equation (7) and the following terms, there are several terms needed to calculate maximization over the entire state space. I don't think this surrogate function is computationally feasible when handling continuous control tasks, e.g., MuJoCo. This is my largest concern for this work. If the authors can explain this properly, I am happy to increase the score.
* Some points about clarity:
  * The meanings of the x-axis and y-axis are missing in Figure 1.
  * Typo: Two w_1 at the end of page 5.

**Questions:**

Please see the details in "weakness". And I have one additional question. I am curious about the results in Figure 1. In general, PPO should have a similar performance compared with TRPO for Ant and Walker. [See https://spinningup.openai.com/en/latest/spinningup/bench.html]. Could you please explain why PPO performs much worse in Figure 1 compared with TRPO?

---

### Official Review · Reviewer_Hx6X · 2023-10-31

**Soundness:** 2 fair
**Presentation:** 1 poor
**Contribution:** 2 fair
**Rating:** 3
**Confidence:** 4

**Summary:**

This work studies on-policy policy optimization with worst-case policy performance as objective rather than conventional expected performance. Absolute MDP is introduced with a formal optimization objective derived from Absolute Performance Bound and Chebyshev’s inequality. This paper then presents a monotonic improvement algorithm by introducing the lower bound of performance expectation and the upper bound of performance variance. By introducing several approximation and surrogates, a practical on-policy algorithm called Absolute Policy Optimization (APO) is proposed. The proposed algorithm is evaluated in both continuous domain and the discrete domain, against A2C, TRPO and PPO.

**Strengths:**

- The motivation is clear. I appreciate the authors’ effort in studying worst-case policy optimization as it is of great significance in practical problems.
- The paper is well organized. Overall, it is easy to follow the main thoughts of this work from formulation, lower/upper bound approximation, practical relaxation and implementation.
- The experiments are diverse, including both continuous control and discrete-action environments.

**Weaknesses:**

I think the presentation of Section 4 needs to be improved substantially. It is not easy to read in current version. The lower/upper bound surrogates are introduced in Equation 4 and all definitions of associated terms are put below in a whole. I recommend the authors to take other means like using a table or list for more compact and clearer presentation. In addition, I think the explanation on the key steps of derivation is lacking.

For Equation 13, I do not see how the weighted sum form addresses the challenge mentioned above.

&nbsp;

For some other thoughts, the optimization objective of AMDP given by Equation 3 reminds me of distributional RL. In distributional RL, the variance of policy performance can be computed with proper presentation of value distribution. Thus it seems that Equation 3 can also be optimized via distributional RL. And intuitively, it is more straightforward to resort to distributional RL when consider the distribution and the worst case of policy performance, especially when approximation and surrogates are inevitable to adopt and the theoretical guarantee may be last. It will be interesting to add some discussion on this.

&nbsp;

For the experiments, the overall advantage of APO is not prominent due to many overlaps in error bars (i.e., the shades). As to the key hyperparameter $k$, the ablation in Section 7.6 is only conducted in one environment, which is not convincing enough to me.

Moreover, I recommend the authors to consider including V-MPO (Song et al., ICLR 2020) as a baseline since it is also a popular and performant on-policy policy optimization algorithm.

&nbsp;


Minors:
- It should be $\omega_1, \omega_2$ in ‘where $\omega_1, \omega_1$ are weights to control the importance of optimization objectives’.
- $\pi_k$ is missing below ‘via importance sampling with a sampling distribution $\pi_k$’. It should be $\pi_j$ I guess.

**Questions:**

1) How can the weighted sum form (Equation 13) addresses the challenges of step size and infeasible solution?

2) How many independent random trials are used for the experiments?

3) How is the computational cost of APO (e.g., wall-clock time)?

---

### Official Review · Reviewer_qB7m · 2023-10-31

**Soundness:** 2 fair
**Presentation:** 2 fair
**Contribution:** 2 fair
**Rating:** 3
**Confidence:** 4

**Summary:**

This paper considers the worst-case performance improvement in policy optimization. It proposes the Absolute Performance Bound as a measure for the policy performance. The paper then shows such bound can be effectively obtained by solving a Chebyshev’s-inequality-inspired optimization problem. The paper further discusses that this optimizaiton problem can be approximately via trust region optimization, which yields a practical algorithm APO. Finally, the paper evaluates the APO against TRPO with empirical experiments.

**Strengths:**

### Novelty and originality
The definition of Absolute Performance Bound is novel and inspiring. This definition seeks to guarantee the performance with a high probability, which can be applied to many safety-concerned scenarios.

The problem formulation of reinforcement learning as to improve “the worst-case performance samples originating from the distribution of the variable” shows the originality. It sets a new perspective to re-think the optimization objective in RL, and can possibly lead to many new RL algorithms.

**Weaknesses:**

### Clarity
**Some important concepts are presented in a rather vague and confusing way**.
1. **One of the most important contribution, monotonic improvement in Theorem 1, is presented in a vague way**. It is unclear how the monotonic improvement has been achieved. In the proof, Equation (76) shows the performance bound can be lower bounded by the differences $M_j(\pi_{j+1}) – M_j(\pi_j)$. However, $M_j(\pi_{j+1})$ itself is a function over $\pi_{j+1}$ and it remain unknowns whether there exists such $\pi_{j+1}$ that leads to $M_j(\pi_{j+1}) > M_j(\pi_j)$. Hence, that “by maximizing $M_j$ at each iteration” will “guarantee that the true absolute performance $B_k$ is non-decreasing” may not hold. The paper needs to show, either via proof or a concrete example, that there exists at least one feasible $\pi_{j+1}$ that yields $M_j(\pi_{j+1}) > M_j(\pi_j)$.

2. **Many conversions and transformations in Optimized APO are not well theoretically justified**. Specifically, the optimization problems in Equation (11) and Equation (12) are inherently different: how the trust region constraint suddenly appears in Equation (12) and how this KL constrained optimization is approximately equivalent to lower bound surrogate-constrained optimization? Furthermore, why “equation 12 may often lead to infeasible solutions”? If this new formulation, as an approximation to the Equation (11), is infeasible, then why do we need this conversion in the first place? Moreover, the optimization in Equation (13) may not even share the same solution to the optimization in Equation (12). Also, the approximation to this Equation (13) in Equation (14) is not even consistent: how the advantage function becomes a squared function?



### Significance
1. **There is a substantial discrepancy between the proposed theory and the devised algorithm**.  While the formulation of improving the worst-case performance samples is interesting and has been theoretically analysed, the resulting algorithm, APO, deviates greatly from the original formulation. Specifically, the original formulation of the problem in Equation (2) & (3), i.e., Chebyshev-motivated variance analysis, is inherently different from the final optimization problem in Equation (13), Trust Region Optimization. This noticeable discrepancy between the original theory and the algorithm APO may invalidate all the theoretical analysis and conclusions for APO.

2. **The final algorithm to the theoretical results is rather similar to the TRPO. Even the implementation shares the same idea to TRPO**. The paper starts with a new set of theory, which tries to investigate the worst-performance sample. It is followed by some in-depth analysis. However, it suddenly ends up with many approximations and the final algorithm shares rather similar ideas to TRPO. I’m wondering how the new set of theory really helps the development of the new method (or they are just a different interpretation of TRPO?). Further, the final KL constrained optimization problem is solved using exactly the same method as in TRPO: second-order approximation to KL.

3. **The empirical results are not sufficient to show the good performance of APO**. Most empirical results in Figure 5 & 6 show that APO performs rather similarly to TRPO, which makes sense to me since these two algorithms have a rather similar formulation. More empirical investigations are needed to support that APO can be better.

**Questions:**

1. How is the weighted sum formulation in Equation (13) derived? I don’t see the direct link between the optimization problems in Equation (13) and Equation (12). Are they equivalent?

2. “approximating the objective via ﬁrst-order expansions, and the trust region constraint via second-order expansions” what is the first order expansion of the objective? Why do you approximate the first order?

Language and formatting issues (which do not affect my assessment):
1. Some of citation formats are not correct (misuse of citep & citet), e.g., TRPO, PPO, (Yuan et al.), (Tomczak et al., 2019) in related work.

2. inconsistent notations: $\mu: \mathcal{S}\mapsto\mathbb{R}$ (a mapping from state to distribution), $\pi: \mathcal{S}\mapsto\mathcal{P}(\mathcal{A})$ (also a mapping from state to distribution)

3. “Markov Decision Processes (MDP) characterized by improvement of the absolute absolute performance.”

4. The variance of the performance distribution $\mathcal{V}$ is defined using the same letter as the value $V$. Would be more intuitive to use $Var(\pi)$